# Potential Applications and Ethical Considerations for Artificial Intelligence in Traumatic Brain Injury Management

**DOI:** 10.3390/biomedicines12112459

**Published:** 2024-10-26

**Authors:** Kryshawna Beard, Ashley M. Pennington, Amina K. Gauff, Kelsey Mitchell, Johanna Smith, Donald W. Marion

**Affiliations:** 1Traumatic Brain Injury Center of Excellence, Silver Spring, MD 20910, USAdonald.w.marion.ctr@health.mil (D.W.M.); 2General Dynamics Information Technology Fairfax Inc., Falls Church, VA 22042, USA; 3Xynergie Federal, LLC, San Juan 00936, Puerto Rico; 4Ciconix, LLC, Annapolis, MD 21401, USA

**Keywords:** TBI, artificial intelligence, machine learning, diagnostic, neuromonitoring

## Abstract

Artificial intelligence (AI) systems have emerged as promising tools for rapidly identifying patterns in large amounts of healthcare data to help guide clinical decision making, as well as to assist with medical education and the planning of research studies. Accumulating evidence suggests AI techniques may be particularly useful for aiding the diagnosis and clinical management of traumatic brain injury (TBI)—a considerably heterogeneous neurologic condition that can be challenging to detect and treat. However, important methodological and ethical concerns with the use of AI in medicine necessitate close monitoring and regulation of these techniques as advancements continue. The purpose of this narrative review is to provide an overview of common AI techniques in medical research and describe recent studies on the possible clinical applications of AI in the context of TBI. Finally, the review describes the ethical challenges with the use of AI in medicine, as well as guidelines from the White House, the Department of Defense (DOD), the National Academies of Sciences, Engineering, and Medicine (NASEM), and other organizations on the appropriate uses of AI in research.

## 1. Introduction

Artificial intelligence (AI) is often praised for its potential to transform many aspects of modern life, resulting in increasing global investment in its advancement and implementation. Due to this enthusiasm, AI-driven tools have already become an integral part of daily life for many people, despite their relatively recent development. AI can be used to perform a wide variety of tasks, such as having human-like conversations, predicting online shoppers’ preferences, powering voice-controlled virtual assistants like Apple’s Siri and Amazon’s Alexa, and deciding which posts to display on social media platforms, such as Facebook and X (formally Twitter) [1]. In medicine, government officials, researchers, and clinicians have shown an increasing interest in expanding the use of AI techniques to help deconvolve complex, multidimensional patient data (e.g., neuroimaging, electroencephalography (EEG), genetic, and blood biomarker data) to improve the speed and accuracy of pathology detection and support clinical decision making [2]. For example, the United Kingdom government has invested GBP 50 million into the establishment of five AI Centres of Excellence in digital medicine and imaging [3]; similarly, the Japanese government has been developing an “AI Hospital System” as a method to support the country’s aging population, mitigate the effects of its diminishing workforce, and improve medical care in rural settings [4].

Traumatic brain injury (TBI) is one example of a neurological condition for which the application of AI has been explored [5,6]. TBIs are particularly common among military personnel, veterans, survivors of intimate partner violence, athletes, and the elderly [7,8,9,10], but they can affect individuals of all sociodemographic backgrounds and occupations and can be sustained through a variety of different mechanisms (e.g., motor vehicle accidents, falls, assaults, or injurious blast exposure). Thus, TBI is considerably heterogeneous, which makes the condition particularly difficult to diagnose and treat [11]. TBIs are currently categorized based on severity as mild, moderate, or severe using tools such as the Glasgow Coma Scale (GCS) in combination with neuroimaging results and the presence and duration of various clinical signs [12,13]. The majority of TBIs are classified as mild, which is characterized by a GCS score of 13–15, a confused or disoriented state lasting less than 24 h, a loss of consciousness for up to 30 min, and/or memory loss lasting less than 24 h based on criteria defined by the American Congress of Rehabilitation Medicine (ACRM), which are also used by the United States Department of Defense (DOD) [12,13]. Moderate and severe TBIs are characterized by GCS score ranges of 9–12 and <9, respectively, and typically involve longer durations of loss of consciousness, altered consciousness, or memory loss, as well as structural pathology on computed tomography (CT) imaging [12].

While these criteria are commonly used to aid triage decisions during the initial evaluation of an individual with suspected TBI, the severity of one’s injury is often an insufficient predictor of the long-term outcome; even among those with mild TBI (mTBI), symptoms including somatic, cognitive, and emotional/behavioral issues have been shown to persist for months to years after the initial injury in 10–25% [14]. Current tools for evaluating TBI have a limited ability to discriminate those who will develop persisting TBI sequelae from those who will recover, since these individuals may exhibit a normal GCS score and no evidence of pathology on neuroimaging [15,16]. Tools like the GCS and CT imaging may provide more useful indicators of prognosis in individuals with moderate to severe TBI, but these tools are limited in their ability to reliably inform treatment decisions beyond surgical and other emergency interventions [17]. Individuals with moderate to severe TBI may present with a variety of structural pathologies, such as hemorrhage, diffuse punctate axonal injury, or diffuse brain swelling, that contribute differently to patient recovery and involve distinct underlying mechanisms [17,18]. The development of personalized therapeutic strategies for TBI has likely been limited because most clinical trials for TBI do not account for this heterogeneity in underlying pathological findings [17]. Thus, variability in TBI outcome and presentation has led to important challenges with its acute and long-term management.

To help refine TBI characterization and eventually enable personalized medicine for TBI, organizations such as the National Academies of Sciences, Engineering, and Medicine (NASEM) and the National Institute of Neurological Disorders and Stroke (NINDS) have endorsed increasing the collection of multimodal clinical data during initial TBI assessment and across multiple timepoints following the initial injury [19,20]. For example, an increasing number of studies have evaluated the clinical value of molecular biomarkers isolated from biofluids, including markers of brain cell damage or inflammation, in the assessment of TBI [15]. This research eventually led to the 2018 clearance of the first blood-based biomarker assay for aiding the evaluation of TBI by the United States Food and Drug Administration (FDA) [21]. Certain eye-tracking devices and neurocognitive assessment tools (e.g., the Automated Neuropsychological Assessment Metrics [ANAM] and Immediate Post-Concussion Assessment and Cognitive Testing [ImPACT]) have also recently received FDA clearance for aiding in the evaluation of TBI, and the latter have become essential components of mTBI follow-up protocols and return-to-activity decisions after mTBI in sports and the military [22,23,24]. In addition, more advanced neuroimaging methods, such as magnetic resonance imaging (MRI) modalities, have started to become incorporated into clinical recommendations for the evaluation of TBI to enable the detection of subtle structural and functional pathologies, thus addressing the limitations of standard CT imaging [25]. Other tools that have been increasingly applied to TBI evaluation in recent studies include EEG, genetic markers, markers of autonomic nervous system dysfunction, and vestibular function assessments [15]. With further study, the combined assessment of these measures with sociodemographic information and patient medical history could substantially improve the clinical management of TBI and enable more personalized treatment options.

These efforts, combined with the development of TBI data repositories like the Federal Interagency Traumatic Brain Injury Research (FITBIR) informatics system indicate that clinicians and researchers will soon have access to the largest amount of data on TBI patients ever to exist [26]. Thus, some researchers have proposed the conceptualization of TBI as a “Big Data” problem, which demands more sophisticated and robust methods to help interpret this rapidly increasing volume and variety of TBI patient data [27]. Due to the complexity of these data and their extensive variability across TBI patients, it is unlikely that a single measure or tool will be able to diagnose or prognosticate TBI with sufficient accuracy or sensitivity. Additionally, because many of the underlying mechanisms and the patient and injury factors contributing to TBI progression remain unclear, it may be difficult to logically design an appropriate panel of measures for predicting the development of specific TBI symptoms or a patient’s response to a particular treatment [28]. Advanced algorithmic methods such as AI techniques have promise for addressing these challenges by quickly and efficiently reducing a large amount of patient data into clinically meaningful information with minimal human supervision [27]. For example, researchers have begun to investigate the application of AI in supporting triage decisions in emergency department settings [29], diagnosing TBI [5], identifying TBI clusters [30], and predicting post-TBI symptoms [31]. Additionally, AI techniques can allow clinical investigators to focus on the creative aspects of developing new research studies equipped with concise summaries of the results of all related published studies.

However, AI also presents new opportunities for the misuse of data and plagiarism [32,33,34] and can demonstrate methodological flaws with improper use, so as the technology improves there is an urgent need for some form of government regulation and guidance regarding the legitimate and appropriate use of AI. One concern is ensuring that AI-assisted clinical tools adhere to patient privacy and security standards as advancements continue. For example, large language models (LLMs) have potential applications in cybersecurity by quickly performing data encryption and automated threat detection with minimal human supervision; however, as cybersecurity threats grow more sophisticated and complex, efforts must be made to develop provisions that prevent malicious actors from compromising patients’ personal information and biometric data [35,36]. Additionally, the lack of transparency and explainability in AI systems have contributed to widespread criticism of these tools and to concerns about whether informed consent can be sufficiently obtained from patients whose data are subjected to AI-based analyses [37,38]. Other crucial ethical considerations include minimizing algorithm biases and ensuring that human oversight is maintained to monitor the safety and efficacy of AI-based medical tools [38].

Due to increasing interest in the use of AI to assist in clinical decision making for TBI evaluation and management and growing appreciation for the ethical concerns with these methods, the purpose of this narrative review is to (1) provide working definitions for various forms of AI; (2) describe evidence on the use of AI systems in TBI evaluation and management; and (3) discuss the ethical challenges with implementing AI systems in healthcare applications and the policies that have been developed to guide these efforts.

## 2. Search Method

PubMed was utilized to conduct the literature search for this narrative review. The search used terms related to artificial intelligence (“artificial intelligence”, “machine learning”, “deep learning”, “large language models”, “reinforcement learning”, “artificial superintelligence”, “artificial general intelligence”, “artificial narrow intelligence”, “predictive artificial intelligence”, “explainable artificial intelligence”, “artificial narrow intelligence”, “SHAP”), terms related to evaluation (“diagnostic”, “prognostic”, “clinical”, “nonclinical”), other related terms (“neuroimaging”, “evaluate”, “diagnosis”, “assess”, “etiology”, “comorbid”, “treatment”, “risk”, “risk factor”, “prevalence”, “protective factors”), and terms related to TBI (“head injuries”, closed”, “brain injuries”, “traumatic”, “brain concussion”, “post-concussion”, “traumatic brain injury”, “TBI”, “mTBI”, “concuss”, “brain injuries”). Articles were retrieved and evaluated for relevance to the subject. References within the identified articles were also searched, and relevant articles were retrieved.

## 3. AI Techniques Commonly Used in Medical Research

**AI** refers to technologies that involve sophisticated algorithms that learn from vast amounts of data to solve problems and perform tasks in ways that would typically require human intervention (Figure 1) [39]. AI systems can identify patterns within data to make predictions, and the incorporation of self-correcting abilities enables an AI system to improve its accuracy through feedback [2]. AI comprises a variety of computational techniques that can be selected based on the dimensionality of the data and specific application [28]. There is a growing impact of AI in healthcare, particularly in diagnostics and personalized medicine. AI systems, such as deep learning models, are achieving human-level performance in areas like medical imaging and disease detection [40]. Despite its promise, AI faces challenges such as data quality, model transparency, and ethical concerns. However, AI has the potential to reduce healthcare costs, improve patient outcomes, and increase access to care [40].

**Artificial narrow intelligence (ANI)**, also referred to as weak AI or narrow AI, has been the most widely utilized and the only commercialized form of AI to date. Narrow AI is goal-oriented and designed to perform singular tasks, e.g., facial recognition, speech recognition, driving a car, or searching the internet—and is very intelligent at completing the specific task it is programmed to do [1]. ANI operates under a narrow set of constraints to perform tasks that simulate human behavior in real-time [1]. One area of medicine for which ANI is highly applicable is imaging analysis, as ANI systems are found to perform similar to or better than radiologists in detecting diseases and abnormalities [41]. However, ANI is only highly proficient in the specialized task it is designed for, meaning it cannot be generalized across broader medical contexts [41].

**Predictive AI** refers to a computer program’s ability to use statistical analysis to identify patterns, anticipate behaviors, and forecast future events [42]. Predictive AI goes beyond simple observations; it analyzes thousands of factors and large volumes of data to inform predictions [42]. Predictive AI thus analyzes vast quantities of data—often referred to as “Big Data”, and the more data available, the better the predictions. AI-driven predictive analytics can improve patient care by predicting health outcomes, disease progression, and response to treatment [43]. With these systems, vast amounts of patient data can be used to provide personalized decision support, allowing healthcare providers to make more informed and timely clinical decisions [43]. By identifying patterns and risks early, predictive AI helps improve patient outcomes and optimize resource allocation in healthcare [43]. Most AI models that have been investigated for healthcare purposes, such as diagnostics and outcome prediction, involve predictive AI.

**Internet of Things (IoT)** refers to a network of physical devices, vehicles, appliances, and other objects equipped with sensors, software, and network connectivity [44]. These smart devices can collect and share data, enabling them to communicate with each other and with other internet-enabled devices. IoT devices, also known as “smart objects”, include simple home devices (like smart thermostats), as well as wearables (such as smartwatches) and complex industrial machinery [45]. The integration of AI into IoT devices can improve the efficiency and accuracy of patient data management. In smart healthcare systems, IoT devices continuously collect patient data in real-time, which can then be analyzed using hybrid AI techniques to optimize diagnostic processes and treatment decisions [46]. The use of these devices can not only enhance data processing but also improve the overall performance of healthcare systems by providing timely and personalized care, leading to more efficient resource allocation and improved patient outcomes [46].

**Machine learning** is a subfield of AI that uses algorithms trained on datasets to create models that can be used to make predictions from subsequent data without explicit instructions or human intervention [28]. Machine learning has been widely investigated for its potential role in predicting disease progression and individual responses to treatments and for its potential utility to optimize treatment strategies [47]. With successful optimization, machine learning strategies can allow healthcare providers to anticipate disease trajectories, improving clinical decision making and enabling more tailored treatment plans that improve patient outcomes [47]. Generally, machine learning can be categorized as supervised or unsupervised learning. With supervised machine learning, an algorithm is trained using data for which the state of the data is known (e.g., each input corresponds to a “TBI patient” or “healthy control”), and the developed model is used to make predictions on new data for which this information is not known [28]. Unsupervised machine learning involves using algorithms to identify patterns in unlabeled data [28]. Common forms of machine learning algorithms used in medical research include the naïve Bayes model, k-Nearest Neighbors, Least Absolute Shrinkage and Selection Operator, random forest, and support vector machine (SVM) [28].

**Deep learning** is a subfield of machine learning that leverages neural networks to perform representation learning, during which the algorithm develops its own representations for pattern recognition after receiving raw data as the input [48]. The term “deep” refers to the use of multiple layers in the network [49]. Deep learning models are based on neural networks, which are inspired by the human brain. These networks consist of interconnected nodes (neurons) that process and transform data [50].

Deep learning aims to transform input data into abstract and composite representations. For example, in an image recognition model, the initial layer identifies basic shapes (like lines and circles), subsequent layers encode more complex features (such as facial features), and the final layer recognizes the overall object (like a face). Unlike traditional machine learning, in which features are manually selected and incorporated into a model, deep learning models automatically learn useful feature representations from the data itself [48]. Deep learning has been successfully applied to various fields, including computer vision (image recognition, object detection, and segmentation), natural language processing (text analysis, sentiment analysis, and language translation), speech recognition (converting spoken language into text), and bioinformatics, including medical image analysis. Deep learning models like convolutional neural networks improve diagnostic capabilities by identifying patterns and abnormalities that might be difficult to detect using traditional methods [51]. Thus, deep learning has the potential to enhance diagnostic accuracy by streamlining imaging analysis, reducing clinician workload, and allowing for faster, more reliable medical decisions [51].

**Reinforcement learning** refers to machine learning techniques that aim to train computational agents to achieve specific functions, and this learning can involve trial and error, expert demonstration, or a hybrid strategy [48]. With reinforcement learning, feedback loops of rewards and penalties are used to make the agent better at accomplishing its designed task [48]. When the model performs actions that move it further away from completing its objective, a penalty is mathematically applied to the model, while actions that move the model closer to achieving its objective are positively reinforced with rewards, which have the opposite sign of penalties [52]. For example, penalties can be assigned negative values and rewards can be given positive values. Robotic-assisted surgery is one healthcare application for which reinforcement learning approaches have been studied [53]. Reinforcement learning can also be applied to the optimization of medication dosages [54]. These algorithms can be integrated with pharmacokinetic/pharmacodynamic models to offer dosage recommendations that are dynamically adjusted based on patient data, with the goal of maximizing therapeutic effects and minimizing side effects [54]. This data-driven approach has the potential to improve treatment efficacy across many medical conditions such as cancer, diabetes, and neurologic diseases, for which the precision of treatment is most critical to outcome [54].

**Generative AI (GAI)** refers to AI systems that can create new information from previously learned data [42]. These systems “invent” new content, solutions, or concepts that did not previously exist in the data they were trained with, rather than merely analyzing or processing the data [55]. GAI systems involve algorithms that are designed to learn from large bodies of information, including scientific literature. Programs like ChatGPT and Midjourney fall under the category of GAI. These programs learn from data (such as online text and images) to create new content that feels human-made. Chatbots like ChatGPT engage in text conversations, while Midjourney generates images from simple text instructions [42]. GAI can be used for medical applications to provide new insights into complex biological processes. In a recently published study, GAI was used to describe brain amyloid load, dynamics, and progression using a cohort of 1259 subjects’ AV45 positron emission tomography (PET) images [56]. This methodology can provide invaluable information for understanding Alzheimer’s disease and could potentially inform its diagnosis and future clinical trial design.

**Natural Language Processing (NLP)** is a subfield of AI that focuses on the interactions between systems and human language. It involves the development of algorithms and models to understand, interpret generate, and respond to human language in a meaningful way [57]. It encompasses tasks such as text classification, sentiment analysis, machine translation, speech recognition, and language generation [57]. NLP often relies on machine learning techniques to accomplish these tasks. Some NLP systems include the original models of virtual assistants such as Siri or Alexa, and language translation services such as Google Translate. NLP can play a pivotal role in extracting valuable insights from unstructured data within electronic health records [58]. NLP techniques, particularly those powered by neural networks, are capable of analyzing vast amounts of clinical text to identify patterns, trends, and relevant medical information that might otherwise be overlooked [58]. These tools could also enhance the efficiency of clinical workflows by interpreting free-text data.

**LLMs** are a category of foundation model trained on immense amounts of data using deep learning algorithms designed to comprehend and generate human language text [59]. These models are a form of GAI that is based on transformer architectures and involve billions of parameters [60]. Examples of LLMs include ChatGPT-3 and GPT-4, Google’s BERT/RoBERTa and PaLM models, and IBM’s Watson Assistant and Watson Orchestrate. These models have become increasingly popular due to their ability to understand and generate natural language, enabling them to perform a wide range of tasks. For example, LLMs can summarize lengthy text into concise versions, answer general questions, and assist in tasks like creative writing and code generation among other tasks. In the context of medicine, these models, trained on vast amounts of medical literature and clinical data, excel at understanding and generating natural language, enabling them to assist in tasks such as summarizing medical documents, generating patient reports, and even aiding in clinical decision making [61]. LLMs can process and interpret complex medical information, making them valuable tools for clinicians and researchers alike [61]. However, there needs to be careful consideration of biases and an emphasis on ensuring transparency to maintain trustworthiness when using LLMs in healthcare applications.

**Explainable AI (XAI)** refers to the ability of an AI system or model to provide clear and understandable explanations for its actions or decisions [62]. AI techniques like deep learning involve the detection of complex, nonlinear relationships in data that are difficult to understand, leading some to refer to these strategies as a “black box” [16]. XAI aims to make AI processes more transparent and interpretable to humans and allows users to comprehend how decisions are made. Within XAI, knowledge graphs are often utilized to increase the explainability within a model, particularly in scenarios that require the output to be linked to well-established domain knowledge, such as in healthcare and biomedical research settings [63]. In the case of biomedical ontologies, knowledge maps include relationships between diseases, treatments, and biological processes, such as with gene ontology or disease ontology [63]. The integration of AI into clinical workflows requires a high degree of interpretability. In the case of knowledge graphs, complex datasets are mapped into structured, interconnected representations to provide an intuitive framework to explain AI decision making [63]. By providing insights into how AI models make decisions, XAI techniques like saliency maps help clinicians understand the rationale behind a model’s outputs [64].

**SHapley Additive exPlanations (SHAP)** maps provide a graphical framework used to interpret the output of machine learning models and are thus considered one type of XAI [65]. SHAP utilizes what are known as Shapley values from cooperative game theory, which provide post hoc measures of how each feature (e.g., age, sex, blood pressure, etc.) contributes to predictions in a machine learning model [66]. This approach breaks down and attributes the impact of individual features, making complex models more transparent and interpretable [66]. In healthcare, SHAP can help clinicians and researchers identify which factors are most critical to diagnosis or treatment outcomes [67]. This approach ensures that AI outputs are accurate, interpretable, and actionable, all of which are essential for integration into clinical practice.

It is important to note that often a given AI system will often involve a combination of different AI technologies. These combinations can improve core functionality, advance capabilities, or increase transparency. For example, the ChatGPT NLP promotes its core functionality, enabling the system to interpret and generate human language and is supported by machine learning models that employ deep learning architecture [68]. The use of LLMs further enhances a system’s capacity for nuanced language processing. This specific use of AI can be classified as ANI, as it operates in a specialized domain. Additionally, XAI techniques, such as SHAP, may be integrated into an AI system to ensure transparency [69].

Hypothetical forms of AI include **artificial general intelligence (AGI)** and **artificial superintelligence (ASI)**. AGI refers to a type of AI that matches or surpasses human capabilities across a range of cognitive tasks [70]. Unlike narrow AI, which is designed for specific tasks, AGI aims to exhibit human-like intelligence and adaptability in various domains capable of cross-domain learning and reasoning. While ANI systems excel at specialized tasks such as image recognition, language translation, or playing chess, AGI would be more versatile and capable of handling novel situations. Achieving AGI remains an ongoing challenge in AI research, as it requires developing algorithms and models that can generalize knowledge, reason abstractly, and learn from limited data [71]. In summary, AGI represents the pursuit of creating AI systems that can think, learn, and adapt like humans, making it a significant milestone in the field of AI.

ASI is a hypothetical software-based AI system with an intellectual scope beyond human intelligence [72]. Essentially, ASI would possess highly advanced thinking skills and surpass human intelligence and cognitive abilities in virtually all domains, including problem solving, creativity, and learning. However, it remains theoretical and has not been achieved yet. Developing ASI requires further advancements in technologies, such as LLMs and multisensory AI, as well as more complex neural networks [72]. According to some experts, the emergence of an ASI is highly unlikely in the near future based on current computer architectures, primarily due to energy constraints, because the amount of energy consumed by a hypothetical ASI system would likely exceed that used to power highly industrialized nations [73].

## 4. Potential Applications of AI in TBI Clinical Care

### 4.1. Applications of AI in Medicine

AI techniques have potential applications in diverse areas of evidence-based medicine, from aiding medical data management to assisting pharmacovigilance efforts and guiding personalized treatment [74,75]. GAI, for example, can be used to guide medical research planning by quickly answering queries and providing a summary of next steps for consideration (Figure 2). Additionally, AI has utility in medical education as it can be used to train students in disciplines that heavily rely on neuroimaging, such as radiology, ultrasound, echocardiography, and pathology [39]. In busy acute care settings, AI and natural language processing could aid triage efforts by generating concise and insightful medical summaries based on a patient’s symptoms; this could substantially reduce the time spent examining clinical records and prevent clinicians from missing key information [76,77,78]. Relatedly, AI can be used to mine the large volumes of data contained in electronic records to identify patients with an elevated risk of developing certain illnesses to help select them for preventative interventions [76,79,80]. For example, AI models have high accuracy in predicting the risk of sudden cardiac arrest, allowing clinicians to intervene in a timely manner [81]. Due to these promising findings and the increasing volume of TBI patient data, clinical researchers have also begun to investigate the use of AI to improve TBI management across the continuum of care (Figure 3) [19].

### 4.2. AI-Assisted Monitoring and Management of TBI in Acute Care Settings

In acute care settings, machine learning methods have been investigated as tools for aiding several aspects of moderate to severe TBI management, such as predicting the incidence of urgent medical events [29]. One retrospective study of over 2000 head trauma patients used 18 patient features, including age, sex, systolic blood pressure, loss of consciousness, and pupil abnormalities, to develop machine learning models to discriminate patients with traumatic intracranial hemorrhage (ICH) from those without ICH; the best-performing model had an area under the receiver operating characteristic curve (AUC) of 0.80, sensitivity of 74%, and specificity of 75% in the validation dataset [83]. Additionally, a feature importance analysis identified 5 of the 18 variables (disorientation, high-energy head trauma, head trauma scar, the eye-opening component of the GCS, and pupil abnormality) as most associated with the model’s accuracy [83]. The main limitation of this study was its retrospective design and inappropriate exclusion of patients with missing data. In a recent prospective analysis of 104 patients with moderate to severe TBI who had been randomized to the control group of a clinical trial investigating tranexamic acid for TBI, models comprising machine learning-selected features (AUC = 0.78) performed better than models comprising expert-selected variables (AUC = 0.68) for predicting progression of ICH [84]. While this study did employ internal cross-validation measures, its main limitation was that the results were not validated with an independent validation dataset of TBI patients, indicating a potentially high risk of bias. Additionally, no data on model calibration were presented, which is important for determining how well the model would likely perform on new data [85]. While both studies performed analyses to determine which features were most related to the model’s accuracy, another potential limitation of both these studies is that neither included certain measures related to cardiovascular function, such as anticoagulant use, which could have improved model performance.

Other studies have aimed to develop machine learning models that can be used to identify TBI patients who require surgical intervention. One study of over 2000 moderate to severe TBI patients reported an AUC of 0.81 when using a machine learning model of 15 features to discriminate those who required neurosurgery within 24 h of hospital admission from those who did not [86]. In this study, SHAP analysis indicated the most predictive variables included GCS score, measures of pupillary abnormality, high blood pressure, and low heart rate [86]. One limitation of this study was that it did not employ appropriate internal validation measures, such as cross-validation or bootstrapping, or an independent validation dataset; instead, this study split one dataset, using 80% for model training and 20% for validation. Another study of 200 moderate TBI patients used age, sex, GCS score, and CT findings to develop an SVM-based model to predict the need for surgical intervention, demonstrating an AUC of 0.93 (82% sensitivity, 84% specificity) [87]. While this study utilized fourfold cross-validation to evaluate the developed model, the performance of the model should be confirmed in a larger, external TBI patient cohort.

Several investigators have also used machine learning methods to successfully predict the risk of mortality for severe TBI patients in the emergency room. These studies have identified features such as age, the timing of neurosurgical intervention, and clinical signs as important predictors of post-TBI mortality [31,88,89,90]. Notably, a recent meta-analysis including 15 studies found that machine learning algorithms outperformed traditional regression models in predicting adverse TBI outcomes [31]. However, this analysis noted extensive heterogeneity in the input variables used for model development across studies, with some studies utilizing more variables than are likely feasible to collect in a busy emergency department [31]. This heterogeneity also highlights the need for standardization in the collection of model input data across studies if AI-driven tools for mortality prediction are to be implemented in acute care settings [31]. Other applications for which AI methods have been investigated in the context of acute TBI management include identifying patients at risk of prolonged mechanical ventilation [91], predicting the lengths of hospital stays [92], and determining the need for head CTs in mTBI patients [93]. Collectively, these studies suggest broad uses for AI methods in acute TBI management if methodological concerns can be addressed.

### 4.3. AI-Assisted Diagnosis of mTBI

The validation of biological correlates of injury that can be used to accurately diagnose mTBI presents a unique challenge due to the mild and variable nature of brain pathology associated with mTBI. To increase the sensitivity of mTBI detection, many researchers have developed AI methods for analyzing genetic, EEG, imaging, and biofluid biomarker data from mTBI patients. For example, one group evaluated the use of a deep learning approach to predict mTBI status using DNA methylation data obtained through epigenetic analysis [94]. This study identified four methylation sites that could each be used to accurately discriminate pediatric mTBI patients from healthy controls (AUCs ≥ 0.9–1), and the combined analysis of these sites with clinical data using their deep learning model achieved a sensitivity and specificity of ≥95% [94].

Many other studies have applied machine learning analysis of metrics from EEG recordings and MRI to discriminate mTBI patients from healthy controls with varying accuracy [95,96,97,98,99,100,101,102]. In one small study, the machine learning analysis of EEG data could be used to discriminate mTBI patients from healthy controls with an accuracy of 95% during model training and 70% during model validation [95]. In another study, machine learning analysis of diffusion tensor imaging metrics thought to reflect axonal pathology discriminated 50 mTBI patients from 50 healthy controls with an accuracy of 84% [96]. Other investigators have algorithmically combined functional connectivity measures with regional entropy values, achieving 75% accuracy in discriminating mTBI patients from healthy controls [100]. The primary limitations of these studies include the lack of validation of the developed models in external validation sets, the limited sample sizes of these studies, and the high dimensionality of these data, all of which could challenge the generalizability of the findings to novel datasets.

Machine learning analysis of biofluid biomarker data is a promising technique for leveraging diverse information provided by molecular markers of direct brain cell damage, inflammation, or metabolic processes to detect mTBI. One group used this approach to develop a panel of six metabolites measured in plasma that accurately discriminated athletes with mTBI from controls, and the accuracy of this panel was demonstrated when measured within six hours post-mTBI and two, three, and seven days post-mTBI [103]. Another study used unsupervised clustering analysis of blood biomarker data from athletes and military personnel in the FITBIR database to define 11 biomarker trajectories, two of which were associated with greater risk of loss of consciousness or posttraumatic amnesia at the time of injury [104]. While these findings are promising, an important consideration for the development of biofluid biomarker panels for mTBI detection is ensuring their specificity to TBI; to this end, future studies in this area should aim to include a non-CNS injury control group.

Together, these studies suggest that the use of more advanced methods for quantifying biomarker data could improve the sensitivity of mTBI detection. However, important challenges currently limit the implementation of this approach. First, these types of biomarker data are not yet part of the routine evaluation of mTBI in most clinical settings; thus, efforts to increase the utilization of multimodal mTBI assessments by clinicians are needed. Additionally, while these tools can be designed for ease-of-use, the data must be appropriately formatted to ensure consistency when used across clinicians, meaning clear common data elements must be established and adhered to. Finally, many scientists also express concerns over the interpretability of findings from machine learning models and the relationship between model findings obtained from biomarker data and actual underlying biological mechanisms. These limitations should be investigated and addressed in future studies.

### 4.4. AI-Based Identification of Phenotypic Clusters

Due to the complexity of TBI, many investigators are interested in identifying clinical subtypes of TBI to improve decision making and personalized treatment [105], and machine learning-based clustering methods could be valuable tools for achieving this goal. One study using this approach identified five distinct clusters of mTBI patients using measures of pain, depression, sleep disturbance, fatigue, and anxiety from the Patient-Reported Outcomes Measurement Information System, symptom measures from the ImPACT neurocognitive assessment, and other metrics [30]. These patient subtypes exhibited symptoms ranging from “minimally complex” to “extremely complex”, with complexity defined based on the number and relative severity of symptoms [30]. Another study that used a similar approach also identified five symptom clusters, with some clusters exhibiting few TBI symptoms and others that developed persisting symptoms that significantly impacted social functioning and work productivity [106].

Many other studies have aimed to use unsupervised machine learning approaches to identify TBI phenotypes and patient clusters [107]. One study used unsupervised machine learning to distinguish symptom clusters in 96 individuals with expected sports-related concussion or postconcussive syndrome, which revealed two clusters primarily characterized by the presence or absence of vestibular symptoms [108]. Clustering methods have also been used to group TBI patients based on response to rehabilitation treatments [109], as well as acute injury features and long-term functional outcomes [110]. However, in addition to the lack of external validation of findings from studies in this area, a limitation of these studies is their reliance on subjective self-report measures to identify symptoms; future studies in this area should aim to evaluate whether the addition of objective measures to clustering methods can better inform TBI phenotypes. Nevertheless, these studies collectively demonstrate the potential for machine learning methods to improve personalized medicine for TBI by enabling the identification of patient groups with persisting or severe symptoms that may benefit from targeted treatments [111,112].

### 4.5. AI-Based Detection of Cognitive Impairment

With increasing evidence suggesting a link between TBI and the development of neurodegenerative disorders, especially among individuals who sustain moderate to severe TBIs or multiple TBIs [113], researchers have begun to examine the use of AI methods that can aid the early detection of cognitive impairment. One study used a machine learning approach to demonstrate that features of white matter hyperintensity detected on MRI could be used to accurately predict cognitive impairment in a sample of older individuals aged 47–84 years [114]. Another study used a similar approach to develop machine learning models of brain age using white matter and gray matter features, showing that the predicted difference between the brain age and chronological age of a group of patients with TBIs of any severity was 4.66–5.97 years [115]. Additionally, this age difference could be used to predict cognitive impairment and was correlated with time elapsed since injury, suggesting TBI initiates cumulative brain atrophy over time [115]. Consistent with these findings, a different study that used machine learning analysis of white matter changes within the default mode network showed that geriatric mTBI patients exhibit functional connectivity patterns similar to those in Alzheimer’s disease patients [116]. However, unlike the former two studies, this study did not evaluate the developed model in an independent dataset. Other studies have investigated AI-based analysis of other metrics, such as eye-tracking data, as a strategy for the early detection of Alzheimer’s disease [117], which may also show promise for detecting neurodegenerative disease following TBI with further study. Collectively, these studies indicate that AI may help with the prediction of individuals at risk for developing chronic conditions following TBI.

### 4.6. Considerations Regarding the Use of AI Methods in TBI Research

Despite the promising potential of AI methods such as machine learning to improve the clinical management of TBI, important challenges currently limit the generalizability of findings from these studies and the implementation of AI models for predicting TBI outcomes. First, these studies often exhibit methodology concerns that contribute to a high risk of bias. In one systematic review of nine studies on machine learning strategies for predicting psychosocial outcomes after TBI, every model was found to have a high risk of bias, and none of the studies provided reliable evidence for the predictive performance of the developed models [85]. Several tools have been developed to help assess bias risk during the development of prediction model studies; these include the Prediction model Risk Of Bias ASsessment Tool (PROBAST) [118] and Transparent Reporting of a multivariable prediction model for Individual Prognosis Or Diagnosis (TRIPOD) checklist [119]. However, few studies in this area report having utilized such tools. Additionally, few studies provide a description of an a priori consideration of how appropriate their machine learning analysis was for the dataset (i.e., based on sample size and outcomes of interest) [85].

These studies exhibit other methodological concerns worthy of consideration. The majority of these studies are retrospective, single-center studies; thus, further confirmation of the findings with multicenter prospective cohorts is warranted. Some studies in this area also do not adequately evaluate the developed model or even fail to report statistics on the model’s performance or calibration [84]. Other studies do not provide a detailed description of how missing data were handled or they inappropriately exclude individuals with missing data, rather than utilizing appropriate imputation strategies [83]. Some studies lacked an independent test set of data for external validation of the developed model [84], leading to a high risk of overfitting, which occurs when a model closely mimics and performs well on one dataset (e.g., the training data) but does not generalize to new data [28]. Additionally, some studies in this area did not perform a feature selection step to minimize the inclusion of potentially irrelevant measures from the model or they did not utilize other methods such as SHAP to determine which measures were most informative. This issue poses important challenges to achieving transparency and explainability with AI-assisted clinical tools for TBI, which may prevent their adoption and limit their use by clinicians.

## 5. AI Challenges, Risks, and Policies for Ethical Use

### 5.1. Potential Challenges with the Use of AI in Healthcare

AI has great potential, but there are also reasonable concerns and limitations regarding its use in healthcare settings. First, more complex algorithms, such as deep learning and other machine learning models, typically require high levels of computing power, especially during model training since this step must be performed using considerably large datasets [120]. Fortunately, however, applying established models to new data requires less computational power, and efforts to increase the efficiency of AI models have already started to succeed [120]. Other challenges relate to inaccurate or nonsensical outputs, often referred to as “hallucinations”, and biased content AI may generate [55]. It is not always obvious when a document is AI generated [121], and the integration of GAI tools into medical decision making, and specifically TBI diagnosis or prognosis, risks propagating errors that could lead to inaccurate diagnoses or inappropriate treatment [55].

Cybersecurity threats are another area of concern with the use of AI systems for healthcare applications. The integration of AI with electronic health record systems and other healthcare databases can create points of data vulnerability when strategies for safeguarding protected health information are not adequately considered [122]. In this context, cyberattacks have the potential to become incredibly sophisticated. For example, there is potential for attackers to reconstruct confidential and sensitive medical information for a patient from details such as the patient’s age, sex, medical history information, and lifestyle factors [123]. Some researchers have investigated the use of AI systems as one strategy for advancing scam detection and cyberattack mitigation efforts, with some success. To this end, some studies have utilized LLMs, machine learning, and deep learning methods to improve data encryption and automated threat detection [35]; for example, one study developed a neural network approach to detect deception during telecommunication [36]. However, such tools are not yet widely utilized in healthcare settings and require further investigation.

Another ethical consideration is that the often-limited transparency of AI systems presents important challenges to ensuring patient-centered care. Patient-centered care emphasizes patients as active participants in their own healing and their right to autonomy and control over medical decisions [37]. Shared decision making is integral to patient-centered care and involves conversations between the patient and clinician, during which the clinician informs the patient about the potential risks and benefits of different courses of treatment, while the patient conveys their values, preferences, and priorities for treatment [124]. These discussions rely on a clinician’s ability to understand the information they are relaying and to convey this information accurately and effectively to the patient. Ensuring the explainability of AI models is thus crucial to achieving this goal, but research in this area often instead relies on “black box” methods with unclear clinical interpretations. Increasing reliance on “black box” AI systems during clinical decision making could substantially reduce patient trust in clinicians and negatively impact doctor-patient relationships, especially if the use of such tools is not routinely disclosed [37].

The possibility that AI-based methods could amplify biases in healthcare is yet another important ethical issue. Selection bias in datasets used during model training has been shown to diminish the accuracy of machine learning algorithms when they are applied to individuals with features that are underrepresented in the training data [125]. For example, studies have demonstrated that selection bias during the development of databases used for automatic facial recognition programs has resulted in particularly low accuracy when using these programs to recognize the faces of darker-skinned women [126]. In clinical studies, where individuals from racial and ethnic minority groups and resource limited settings are often underrepresented [127,128], there is potential for a similar form of bias during the development of AI-based clinical tools. Without deliberate efforts to utilize data that adequately and equitably reflects individuals from diverse backgrounds, it is possible that AI-based clinical decision-making tools will exacerbate barriers to healthcare for underrepresented individuals, rather than reducing them.

Finally, there is concern over maintaining human oversight and the need for developing regulatory pathways for monitoring the safety and efficacy of AI-driven clinical products. To this end, the FDA released its Artificial Intelligence/Machine Learning (AI/ML)-Based Software as a Medical Device (SaMD) Action Plan in 2021, which proposes a potential framework for regulating AI-driven clinical products that is based on current pathways for regulating medical devices [129]. However, there are limitations to applying a device-centered approach to regulating AI products. For example, it is unclear what measures would be most appropriate for monitoring the efficacy of an AI product once it has been marketed and is used in the real world [130]. Since some AI-based clinical tools will likely be able to be used with little or no human supervision, some have proposed supplementing a device-centric regulatory strategy with a scheme that considers these tools as “physician extenders”; under this scheme, the FDA review of an AI clinical product would involve consideration for whether a tool could be used independently of physician oversight [130]. This scheme is similar to the oversight of nurse practitioners and has similar potential benefits to allowing nurse practitioners to practice independently, such as increasing healthcare access at a lower cost [130].

In summary, while the accelerating development of AI-driven clinical products is undeniable and their potential benefits are clear, there are substantial ethical challenges to consider. The efficiency and accuracy of AI models must be optimized, and robust cybersecurity systems must be developed to ensure the privacy of patient data can be maintained. Improving the transparency and explainability of AI systems is key to preserving patient-centered care, and strategies for minimizing the risk of bias are required to ensure these tools are used equitably. Collectively, these challenges illustrate the importance of developing mitigation strategies and minimum standards for the ethical use of AI in medicine, as well as pathways for regulatory oversight (Table 1).

### 5.2. Views from the Scientific Community on the Ethical Use of AI

Efforts to develop ethical standards for AI use have intensified in recent years. These efforts have involved the combined contributions of scientists, as well as experts in law, ethics, human rights, and digital technology to establish clear principles for the ethical use of AI in healthcare. In an editorial published by the Proceedings of the National Academy of Sciences, a peer reviewed journal of the National Academy of Sciences, an interdisciplinary group of experts urged the scientific community to follow five principles of human accountability and responsibility when using AI in research [131]: Transparent disclosure and attributionVerification of AI-generated content and analysesDocumentation of AI-generated dataA focus on ethics and equityContinuous monitoring, oversight, and public engagement

For each principle, the authors identify specific actions that should be taken by scientists, those who create models that use AI, and others. For example, for researchers, transparent disclosure and attribution includes steps such as clearly disclosing the use of GAI in research—including the specific tools, algorithms, and settings employed—and accurately attributing the human and AI sources of information or ideas. For model creators and refiners, transparent disclosure and attribution means actions such as providing publicly accessible details about models, including the data used to train or refine them. The editorial emphasizes that advances in GAI represent a transformative moment for science—one that will accelerate scientific discovery, but also challenge core norms and values of science, such as accountability, transparency, equity, replicability, and human responsibility. Of these principles, ensuring the ethical and equitable use of AI in medicine is particularly important to maintain patient-centered care and ensure that current biases in healthcare are not amplified by the incorporation of AI-based clinical decision-making tools.

### 5.3. United States (U.S.) Policies on Ethical Uses of AI

At the national level in the U.S., a White House Executive Order about AI: Policy and Principles [132] was issued in 2023 and emphasizes several key points. First, AI must be safe, secure, and trustworthy, and there must be responsible innovation, competition, and collaboration. Additionally, there should be a commitment to supporting American workers, and uses of AI should be consistent with advancing equity and civil rights. The Executive Order further states that the interests of Americans who increasingly use, interact with, or purchase AI and AI-enabled products in their daily lives must be protected. Relatedly, the Executive Order states that Americans’ privacy and civil liberties must be protected as AI continues advancing. Finally, the Executive Order highlights that is important to manage the risks from the Federal Government’s own use of AI and increase its internal capacity to regulate, govern, and support responsible use of AI to deliver better results for Americans, and that the Federal Government should lead the way to global societal, economic, and technological progress.

The 2022 National Defense Strategy further emphasizes the importance of investing in AI to help build enduring military advantages [137]. To this end, in November 2023, the Chief Digital AI Office of the U.S. DOD released its strategy, known as the 2023 Data, Analytics and Artificial Intelligence Adoption Strategy, to accelerate the adoption of advanced AI capabilities to ensure U.S. warfighters maintain decision superiority, or the ability to make and implement more informed and accurate decisions faster than adversaries, on the battlefield for years to come [133]. These principles apply to both combat and non-combat functions and are intended to assist the U.S. military in upholding legal, ethical, and policy commitments in the field of AI. The department’s AI ethical principles encompass five major areas, including that the use of AI should be (1) responsible, (2) equitable, (3) traceable, (4) reliable, and (5) governable.

The U.S. Department of Health and Human Services (HHS) has also established an AI strategy, which includes establishing an AI Council that will support the governance and development of AI throughout the HHS with four focus areas, which include (1) developing an AI-ready workforce and strengthening AI culture; (2) encouraging health AI innovation and research and development; (3) democratizing foundational AI tools and resources; and (4) promoting trustworthy AI use and development [138]. Similar focus areas for ensuring the responsible use of AI are also emphasized within the Department of Veteran Affairs [139], as well as the military health system as part of its effort to advance its digital strategy [140]. The U.S. has also established a Federal Policy for the Protection of Human Subjects, also known as the “Common Rule” [141], to provide protections for the subjects of human studies that must be considered during the use of AI in clinical research. The use of AI in human studies could pose challenges to adhering with this policy as it becomes more difficult to determine what information is truly private or identifiable and what constitutes informed consent [142]. As research in this area continues, careful consideration of these principles and potential challenges should occur at each stage of a given study.

### 5.4. Global Policies on Ethical Uses of AI

Several international organizations have also developed guidelines for the responsible use of AI. In July of 2024, the North Atlantic Treaty Organization (NATO) released an update to its AI strategy, which describes the desired outcomes for the incorporation of AI strategies, as well as goals for responsibly integrating AI into NATO [134]. The strategy endorses six Principles of Responsible Use for AI in Defense, which include Lawfulness, Responsibility and Accountability, Explainability and Traceability, Reliability, Governability, and Bias Mitigation [134]. Notably, the updated NATO AI strategy differs from the 2021 version, as it specifically addresses the misuse of AI and AI-generated disinformation as important areas of concern that necessitate increasing vigilance [143].

In 2021, the World Health Organization (WHO) released their Guidance on the Ethics and Governance of Artificial Intelligence for Health, which emphasizes that as the use of AI in healthcare continues to progress, consideration for human rights and ethics must be at the forefront of these developments [135]. The WHO identifies six ethical principles for the use of AI in healthcare, including: (1) protect autonomy; (2) promote human well-being, human safety, and public interest; (3) ensure transparency, explainability, and intelligibility; (4) foster responsibility and accountability; (5) ensure inclusiveness and equity; (6) promote artificial intelligence that is responsive and attainable.

Additionally, in 2021, the FDA, Health Canada, and the United Kingdom’s Medicines and Healthcare Products Regulatory Agency (MHRA) developed 10 guiding principles for good machine learning practice (GMLP) [136]. These principles offer specific guidance on how to promote the development of high-quality medical products involving AI and machine learning. For example, principle four highlights that training datasets should be independent of test datasets, while principle eight proposes that AI-driven medical products should be tested in clinically relevant conditions. The principles also address several potential concerns with ethical AI use and maintaining transparency; for instance, principle three emphasizes the necessity of using study participants and data that adequately reflect the intended patient population, and principle nine states that users should be provided clear, essential information on the intended use of AI products.

Collectively, these policies lay a foundation for the ethical use of AI products that can be amended as advancements in AI development continue, and careful adherence to them at an international level will be key to promoting the ethical use of AI systems in medicine.

## 6. Discussion

AI and its subcategories represent a rapidly evolving field with considerable potential for facilitating and enhancing TBI research and clinical decision making with further study. With AI-driven clinical tools, important and clinically relevant conclusions can be produced in a fraction of the time it would take with conventional methods. In the context of the acute management of moderate to severe TBI, several studies have demonstrated the utility of AI in assisting with key aspects of triage, including predicting the development of ICH [83,84], identifying patients who require surgical intervention [86,87], and assessing mortality risk [88,89,90]. To accelerate progress in diagnosing mTBI, some researchers have leveraged machine learning analyses of the rich set of data provided by EEG, MRI, biofluid markers and other measures to accurately discriminate mTBI patients from healthy controls [96,98,103]. AI systems have also shown potential in improving the long-term management of TBI through classifying TBI phenotypic clusters and predicting the development of cognitive impairment [106,108,110,114,115]. Collectively, these studies indicate that AI-driven medical tools could allow clinicians to move beyond nonspecific, severity-based TBI classification into a new era in which patients are selected for targeted treatments and interventions based on their unique presentation of clinical signs, sociodemographic features, symptoms, and pathologic findings.

However, research in this area does exhibit several limitations that will be important to consider as work in this area continues. To develop robust AI systems that can be applied to TBI clinical care, common data elements must be identified and standardized [27]. Studies in this area must also aim to avoid methodological flaws that could undermine the quality of resultant AI products. For example, these studies must include and better describe efforts to minimize the risk of overfitting, such as by using independent training and test datasets of sufficient sample size and removing potentially irrelevant measures from diagnostic panels [85]. One potential barrier to the clinical implementation of AI products for TBI management is that some forms of AI require considerably large amounts of computing power and thus may not be applicable in resource-limited settings. Additionally, the ethical challenges with using AI-based analyses of sensitive patient healthcare data are numerous. There clearly are legitimate ways in which investigators and clinicians can and should consider using AI to improve their clinical investigation, but in all cases, that use should be disclosed, properly attributed, and developed in concert with appropriate published guidelines.

Prioritizing transparency and maintaining patient-centered care are particularly important for researchers and clinicians who aim to develop AI products that can be applied to TBI management. While there are currently no FDA-cleared treatments for TBI, many clinicians still recommend various options for managing long-term TBI symptoms in accordance with clinical practice guidelines. For example, cognitive rehabilitation programs may be beneficial for addressing cognitive deficit after TBI and are often recommended for this purpose [12,144]. Beyond these evidence-based treatments, there are many other potential treatment options ranging from neuromodulation strategies to exercise therapies that have been increasingly investigated for the management of TBI symptoms, with varying levels of evidence regarding their efficacy [145,146]. Patients often rely on medical professionals to help them navigate this complex landscape of potential treatments and make informed decisions about pursuing these alternatives. If AI products become part of the process for selecting patients for specific treatments in instances where the most appropriate treatment is not obvious, it will be important for clinicians to be able to communicate the use of these tools to safeguard patient trust. Adherence to guidelines for the ethical use of AI in healthcare, as well as close surveillance of the use of these tools, will thus be crucial for ensuring AI is used responsibly as the field rapidly continues to evolve.

## Figures and Tables

**Figure 1 biomedicines-12-02459-f001:**
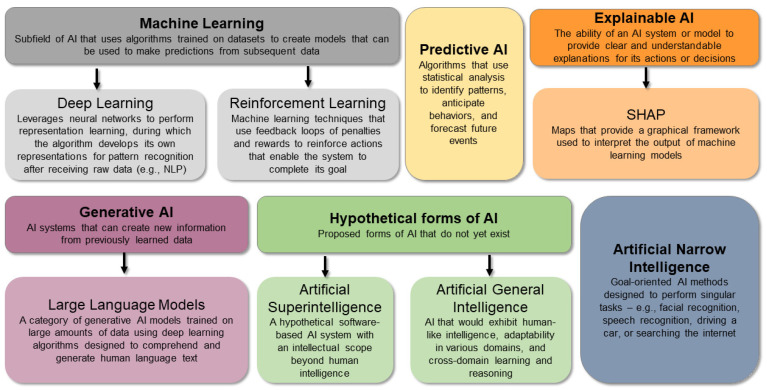
**Artificial intelligence (AI) techniques commonly used in medical research.** AI refers to technologies that involve sophisticated algorithms that learn from vast amounts of data to solve problems and perform tasks in ways that would typically require human intervention. Of note, a given AI system may involve the use of a combination of multiple methods. NLP: Natural Language Processing; SHAP: SHapley Additive exPlanations.

**Figure 2 biomedicines-12-02459-f002:**
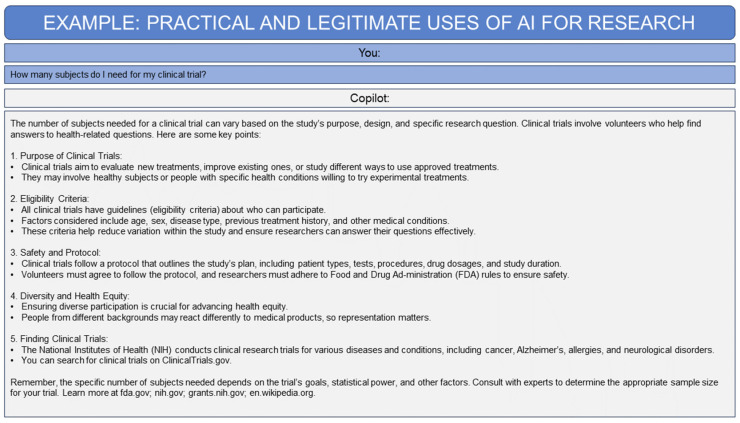
Example use of generative artificial intelligence (GAI) for research. GAI may be useful in guiding research study planning as shown in this example of the reasonable use of Microsoft CoPilot to help in the development of a clinical grant proposal [82].

**Figure 3 biomedicines-12-02459-f003:**
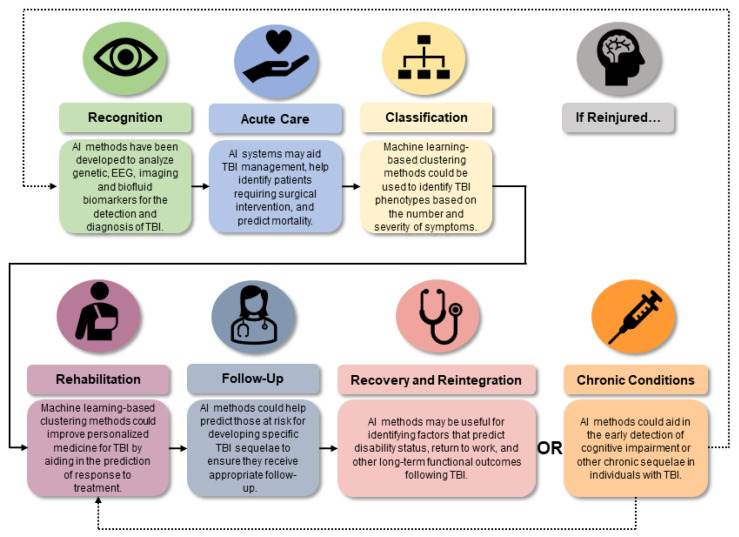
Potential applications of artificial intelligence (AI) throughout the traumatic brain injury (TBI) continuum of care. A recent report from the National Academies of Sciences, Engineering, and Medicine (NASEM) on accelerating progress in TBI care identified key challenges that must be addressed to improve the clinical management of TBI [19]. These challenges can be roughly aligned with specific stages in the continuum of care for TBI, which include its recognition, acute care, classification, rehabilitation, and follow-up, at which recovery and reintegration or the development of chronic conditions can be assessed. Individuals who develop chronic conditions following TBI may undergo rehabilitation to promote their recovery; in the event of reinjury, the continuum begins again with the recognition or detection of the TBI. AI-based techniques have demonstrated the potential to assist with each of these stages in the continuum of TBI care.

**Table 1 biomedicines-12-02459-t001:** Guidelines and policies on the ethical use of artificial intelligence (AI). A number of United States and international organizations have developed guidelines and policies for the ethical uses of AI, many of which are applicable to the use of AI in healthcare settings. The potential risks of AI to propagate biases and encourage the misuse of medical information demand the close surveillance and regulation of AI systems as they continue to develop.

Organization	Guideline/Policy	Policy Principles
Guidelines developed in the scientific community		
National Academy of Sciences	Five Principles of Human Accountability and Responsibility when Using AI in Research [131]	Transparent disclosure and attribution Verification of AI-generated content and analyses Documentation of AI-generated data A focus on ethics and equity Continuous monitoring, oversight, and public engagement
United States government policies and guidelines		
The White House Office	Executive Order about AI: Policy and Principles [132]	AI must be safe, secure, and trustworthy. There must be responsible innovation, competition, and collaboration. There should be a commitment to supporting American workers. Uses of AI should be consistent with advancing equity and civil rights. The interests of Americans who increasingly use, interact with, or purchase AI and AI-enabled products in their daily lives must be protected. Americans’ privacy and civil liberties must be protected as AI continues to advance. It is important to manage the risks from the Federal Government’s own use of AI and increase its internal capacity to regulate, govern, and support responsible use of AI to deliver better results for Americans. The Federal Government should lead the way to global societal, economic, and technological progress.
Department of Defense (DOD)	The 2023 Data, Analytics and Artificial Intelligence Adoption Strategy [133]	Responsible: DOD personnel should exercise appropriate levels of judgment and care, while remaining responsible for the development, deployment, and use of AI capabilities. Equitable: The Department should take deliberate steps to minimize unintended bias in AI capabilities. Traceable: The Department’s AI capabilities should be developed and deployed such that relevant personnel possess an appropriate understanding of the technology, development processes, and operational methods applicable to AI capabilities, including with transparent and auditable methodologies, data sources, and design procedure and documentation. Reliable: The Department’s AI capabilities should have explicit, well-defined uses, and the safety, security, and effectiveness of such capabilities will be subject to testing and assurance within those defined uses across their entire life cycles. Governable: The Department should design and engineer AI capabilities to fulfill their intended functions while possessing the ability to detect and avoid unintended consequences, and the ability to disengage or deactivate deployed systems that demonstrate unintended behavior.
International policies and guidelines		
North Atlantic Treaty Organization (NATO)	Principles of Responsible Use for AI in Defense [134]	Lawfulness, Responsibility, and Accountability Explainability and Traceability Reliability Governability Bias Mitigation
World Health Organization (WHO)	Guidance on the Ethics and Governance of Artificial Intelligence for Health [135]	Protect autonomy. Promote human well-being, human safety, and public interest. Ensure transparency, explainability, and intelligibility. Foster responsibility and accountability. Ensure inclusiveness and equity. Promote artificial intelligence that is responsive and attainable.
United States Food and Drug Administration (FDA), Health Canada, and the United Kingdom Medicines and Healthcare Products Regulatory Agency (MHRA)	Good Machine Learning Practice for Medical Device Development:Guiding Principles [136]	Multidisciplinary expertise is leveraged throughout the total product lifecycle. Good software engineering and security practices are implemented. Clinical study participants and datasets are representative of the intended patient population. Training datasets are independent of test sets. Selected reference datasets are based upon the best available methods. Model design is tailored to the available data and reflects the intended use of the device. Focus is placed on the performance of the human–AI team. Testing demonstrates device performance during clinically relevant conditions. Users are provided with clear, essential information. Deployed models are monitored for performance and retraining risks are managed.

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
