# Peer review of "Potential Applications and Ethical Considerations for Artificial Intelligence in Traumatic Brain Injury Management"

_biomedicines, 2024, doi:10.3390/biomedicines12112459_

Round 1
Reviewer 1 Report
Comments and Suggestions for Authors
An artificial intelligence (AI) systems development as promising tools for rapidly identifying patterns in large amounts of healthcare data to help guide clinical decision making, as well as to assist with medical education and the planning of research studies is actual and relevant to the field. However, despite the significant contribution of the authors, the manuscript does not look sufficiently finalized. Therefore, I can suggest the following directions for enriching the material.
First of all, it is necessary to present the methodology of writing a review, as well as to specify the research question/s. It may be appropriate to add a section on materials and methods, for a more logical transition from the introduction to the body of knowledge.
I understand the emergence of a knowledge gap between new AI approaches and real medical practice, but the manuscript is called potential applications, and I lacked 2024-25 studies and early access, as well as additional specification, which I will make a separate comment on below.
Medical data are not only personal, but also biometric, and the manuscript is practically absent from the aspect of cybersecurity. I propose to supplement the introduction and reference sections with a discussion of the LLM use for these purposes, as well as neural network models for processing biometric data based on:
Pleshakova, E., Osipov, A., Gataullin, S. et al. Next gen cybersecurity paradigm towards artificial general intelligence: Russian market challenges and future global technological trends. J Comput Virol Hack Tech (2024). https://doi.org/10.1007/s11416-024-00529-x
Osipov, A., Pleshakova, E., Liu, Y. et al. Machine learning methods for speech emotion recognition on telecommunication systems. J Comput Virol Hack Tech (2023). https://doi.org/10.1007/s11416-023-00500-2
Frankly, I got the impression that the essential part of the work, as well as the resulting part, is not systematized enough, which makes a comprehensive assessment difficult. Probably, after the task statement is specified and the research methodology is supplemented, this issue will cease to be relevant, but I recommend paying attention to this. The manuscript requires careful author editing in accordance with the rules of design, for example, I am not sure that the word “title” is appropriate in the title.
Strongly hope my comments will be useful to the authors and good luck.
Author Response
Comment 1: An artificial intelligence (AI) systems development as promising tools for rapidly identifying patterns in large amounts of healthcare data to help guide clinical decision making, as well as to assist with medical education and the planning of research studies is actual and relevant to the field. However, despite the significant contribution of the authors, the manuscript does not look sufficiently finalized. Therefore, I can suggest the following directions for enriching the material.
Response 1: We thank the reviewer for this feedback and are pleased with the reviewer’s acknowledgement of the significance of the topic of AI in healthcare.
Comment 2: First of all, it is necessary to present the methodology of writing a review, as well as to specify the research question/s. It may be appropriate to add a section on materials and methods, for a more logical transition from the introduction to the body of knowledge.
Response 2: We appreciate the reviewer for raising these concerns. To address the methodology, we have added a section entitled “SEARCH METHOD” to the revised manuscript as follows (Pages 3-4, Lines 145-158):
- SEARCH METHOD
PubMed was utilized to conduct the literature search for this narrative review. The search used terms related to artificial intelligence (“artificial intelligence”, “machine learning”, “deep learning”, “large language models”, “reinforcement learning”, “artificial superintelligence”, “artificial general intelligence”, “artificial narrow intelligence”, “predictive artificial intelligence”, “explainable artificial intelligence”, “artificial narrow intelligence”, “SHAP”), terms related to evaluation (“diagnostic”, “prognostic”, “clinical”, “nonclinical”), other related terms (“neuroimaging”, “evaluate”, “diagnosis”, “assess”, “etiology”, “comorbid”, “treatment”, “risk”, “risk factor”, “prevalence”, “protective factors”), and terms related to TBI (“head injuries”, closed”, “brain injuries”, “traumatic”, “brain concussion”, “post-concussion”, “traumatic brain injury”, “TBI”, “mTBI”, “concuss”, “brain injuries”). Articles were retrieved and evaluated for relevance to the subject. References within the identified articles were also searched, and relevant articles were retrieved.
To clarify the specific research questions addressed by the review and improve the transition between the INTRODUCTION and the discussion of the evidence, we have added the following to the last paragraph of the INTRODUCTION (Page 3, Lines 139-144):
Due to increasing interest in the use of AI to assist in clinical decision making for TBI evaluation and management and growing appreciation for the ethical concerns with these methods, the purpose of this narrative review is to 1) provide working definitions for various forms of AI; 2) describe evidence on the use of AI systems in TBI evaluation and management; and 3) discuss the ethical challenges with implementing AI systems in healthcare applications and the policies that have been developed to guide these efforts.
To further clarify the scope and purpose of our manuscript, we have changed the Title to the following: Potential Applications and Ethical Considerations for Artificial Intelligence in Traumatic Brain Injury Management
Comment 3: I understand the emergence of a knowledge gap between new AI approaches and real medical practice, but the manuscript is called potential applications, and I lacked 2024-25 studies and early access, as well as additional specification, which I will make a separate comment on below.Medical data are not only personal, but also biometric, and the manuscript is practically absent from the aspect of cybersecurity. I propose to supplement the introduction and reference sections with a discussion of the LLM use for these purposes, as well as neural network models for processing biometric data based on:
-
- Pleshakova, E., Osipov, A., Gataullin, S. et al. Next gen cybersecurity paradigm towards artificial general intelligence: Russian market challenges and future global technological trends. J Comput Virol Hack Tech (2024). https://doi.org/10.1007/s11416-024-00529-x
- Osipov, A., Pleshakova, E., Liu, Y. et al. Machine learning methods for speech emotion recognition on telecommunication systems. J Comput Virol Hack Tech (2023). https://doi.org/10.1007/s11416-023-00500-2
Response 3: We agree with the reviewer that a more comprehensive discussion of the cybersecurity implications of the increasing use of AI in healthcare applications is important. To address this concern, we added the two references mentioned by the reviewer to the INTRODUCTION as follows (Page 3, Lines 127-133):
However, AI also presents new opportunities for misuse of data and plagiarism33-35 and can demonstrate methodological flaws with improper use, so as the technology improves there is an urgent need for some form of government regulation and guidance regarding the legitimate and appropriate use of AI. One concern is ensuring that AI-assisted clinical tools adhere to patient privacy and security standards as advancements continue. For example, large language models (LLMs) have potential applications in cybersecurity by quickly performing data encryption and automated threat detection with minimal human supervision; however, as cybersecurity threats grow more sophisticated and complex, efforts must be made to develop provisions that prevent malicious actors from compromising patients’ personal information and biometric data.35,36
Additionally, we have included a more in-depth discussion of the cybersecurity concerns with AI-driven clinical tools to section 5, AI CHALLENGES, RISKS, AND POLICIES FOR ETHICAL USE, subsection 5.1 (Pages 13-14, Lines 600-613):
Cybersecurity threats are another area of concern with the use of AI systems for healthcare applications. The integration of AI with electronic health record systems and other healthcare databases can create points of data vulnerability when strategies for safeguarding protected health information are not adequately considered.122 In this context, cyberattacks have the potential to become incredibly sophisticated. For example, there is potential for attackers to reconstruct confidential and sensitive medical information on a patient from details such as the patient’s age, sex, medical history information, and lifestyle factors.123 Some researchers have investigated the use of AI systems as one strategy for advancing scam detection and cyberattack mitigation efforts, with some success. To this end, some studies have utilized LLMs, machine learning, and deep learning methods to improve data encryption and automated threat detection;35 for example, one study developed a neural network approach to detect deception during telecommunication.36 However, such tools are not yet widely utilized in healthcare settings and require further investigation.
Comment 4: Frankly, I got the impression that the essential part of the work, as well as the resulting part, is not systematized enough, which makes a comprehensive assessment difficult. Probably, after the task statement is specified and the research methodology is supplemented, this issue will cease to be relevant, but I recommend paying attention to this.
Response 4: We appreciate the reviewer’s feedback and hope the inclusion of the SEARCH METHOD section and our statement of the research question in the INTRODUCTION addresses this concern. Our aim was to develop a narrative review describing research on the use of AI systems for TBI management, as well as relevant ethical considerations for the use of AI in healthcare, rather than to conduct a systematic review or meta-analysis.
Comment 5: The manuscript requires careful author editing in accordance with the rules of design, for example, I am not sure that the word “title” is appropriate in the title.
Response 5: We thank the reviewer for pointing this out and have removed the word “title” from the Title. We have carefully reviewed the manuscript for other potential stylistic issues.
Comment 6: Strongly hope my comments will be useful to the authors and good luck.
Response 6: We greatly appreciate the reviewer’s feedback and thank the reviewer for their contributions to improving the clarity, flow, and depth of our manuscript.

Reviewer 2 Report
Comments and Suggestions for Authors
This paper examines the potential of artificial intelligence (AI) in the management of traumatic brain injury (TBI). Artificial intelligence (AI) shows promise in improving diagnosis, patient outcome predictions, and treatment personalisation, particularly by analysing complex, multidimensional patient data such as neuroimaging and biomarkers. Several AI methods, including machine learning and deep learning, have demonstrated success in detecting intracranial haemorrhages, predicting post-TBI symptoms, and identifying clusters of patients with similar conditions. However, there are challenges, including the need for ethical use, concerns over biases, and computational limitations.
The paper is ambitious in scope and addresses a timely and important problem in a clear manner. However, the paper is somewhat disorganized and lacks sufficient detail to effectively convey the underlying idea. The introduction merely presents a few general statements without adequately introducing the research question. For instance, a literature review is absent or, at the very least, insufficiently developed. This makes it challenging to fully comprehend the author's understanding of AI. Furthermore, the paper fails to address several key topics in the field of AI, including knowledge-graph driven approaches and those derived from biomedical ontologies. Section 3 is well-written and provides a valuable contribution to the discussion; however, it would benefit from a more robust foundation in Section 2. Section 4 is somewhat lacking in this paper. It would be beneficial to the reader if more examples were provided, for example, for each use case in Section 3, or if this example were better embedded.
Section 5 is noteworthy and constitutes a significant contribution. However, the paper lacks depth, particularly with regard to several aspects of ethics. For instance, it does not address patient-centric perspectives or the differences between "classical" data-centric approaches and AI. These are just two examples of areas that could have been explored in greater depth.
The conclusions presented in Section 6 reinforce the shortcomings of the paper. It is not sufficiently detailed to engage with AI as a technology within the context of computer science. Additionally, it lacks sufficient references, examples, and discussion, including on ethical issues.
In light of these concerns, it is recommended that the manuscript undergo further revision.
The English in the text requires improvement to more clearly express the research. It also requires further proofreading.
Round 2
Reviewer 1 Report
Comments and Suggestions for Authors
This version is well written and can be accepted. All comments have been adressed. Thank you.